# Factors Influencing Progressive Utilization of Palliative Care Services among Cancer Patients in Kenya: The Case of Nairobi Hospice

**DOI:** 10.3390/ijerph20196871

**Published:** 2023-10-02

**Authors:** Caroline Wambui Kimani, Urbanus Mutuku Kioko, Catherine Ndinda, Pauline Wambui Adebayo

**Affiliations:** 1School of Economics, Population and Development Studies, University of Nairobi, Nairobi Campus, Nairobi, Kenya; ckimani@kmtc.ac.ke (C.W.K.);; 2Human Sciences Research Council, Pretoria 0001, South Africa; 3Development Studies, University of South Africa, Pretoria 0002, South Africa; 4School of Built Environment & Development Studies, University of Kwazulu-Natal, Durban 4041, South Africa

**Keywords:** progressive utilization, palliative care, cancer patients

## Abstract

The rising cases of non-communicable diseases, specifically cancer, have led to the integration of palliative care in their management. However, only 10% of cancer patients have access to palliative care. Healthcare utilization is an important step in disease management as it aids individuals in accessing opportunities for the prevention and treatment of diseases. The study applied the binary probit model to estimate the progressive utilization of palliative care services by cancer patients. The aim of the study was to determine factors influencing the progressive utilization of palliative care by cancer patients. A cross-sectional data survey was conducted for 169 cancer patients seeking palliative care at the Nairobi Hospice in 2013. For each patient, the predisposing, enabling, and need (PEN) factors were analyzed as key criteria for applying progressive utilization of palliative care at the Nairobi Hospice as compared to those residing in other counties in the study. Descriptive statistics showed that 27% of patients studied resided in Nairobi County, where 61% were female, 62% were married, 35% had primary education, 44% were self-employed, and 59% had medical insurance. Probit regression and marginal effects showed that employment and religion were significant in determining the progressive utilization of palliative care. Employment status and religion are consequently the main factors that both governments and health-focused non-governmental organizations need to consider increasing the probability of progressively utilizing palliative care to improve the quality of life of cancer patients.

## 1. Introduction

Palliative care endeavors to improve the quality of life of patients with life-limiting diseases and their families through the prevention and relief of suffering by means of early identification, correct assessment, and treatment of pain as well as other problems, physical, psychosocial, and spiritual [1]. Globally, approximately 56.8 million people need palliative care services; 78% of them live in low and middle-income countries, yet only about 12% have their needs met, causing great suffering for many. Most adults in need of palliative care have non-communicable diseases, accounting for about 69%.

Worldwide, over 71% of deaths result from non-communicable diseases, including cancer. These diseases result in approximately 9.3 million deaths annually and require palliative care for their management. These statistics are projected to rise, with the greatest increase in low and middle-income countries and communities (LMIC) [2]. The World Health Organization’s (WHO) global action plan 2013 [3] on prevention and control of non-communicable diseases noted with concern that cancer is among the conditions for which millions of patients require palliative care. This means that there is a need for a comprehensive cancer care program to relieve the suffering of the patients while improving their quality of life. Patients receiving palliative care services have shown remarkable improvement in quality of life [4], and as such, the concept has gained global recognition. Due to these perceived benefits, the WHO set standards for palliative care and pain control. These were first applied in developed countries and then extended to the developing world.

Theoretical frameworks for analyzing healthcare utilization are based on three components that include characteristics of the health service delivery system, changes in medical technologies, and individual determinants. Several approaches have been used to study the utilization of healthcare services, including the socio-cultural approach, which views healthcare as part of a cultural complex. In this approach, the organization of hospitals and other healthcare services is based on the cultural setting in the community [5]. The socio-psychological approach, on the other hand, identifies knowledge, belief, and attitude towards symptoms experienced by a patient as influences on decision-making in healthcare seeking [6].

Another approach is socio-demographic, which looks at variations in utilization of healthcare services based on age, gender, education level, occupation, and socioeconomic status, as used in this study. Some of these factors in relation to healthcare utilization have remained stable, but others are deemed to change over time.

The Kenyan healthcare system has majorly leaned towards the public sector, and even following devolution in August 2010 after the inauguration of the new constitution [7], this has remained largely so. The main changes were the transfer of most of the healthcare delivery services to the county governments, except for the National Referral Hospitals. The justification for this was to empower the county governments to exercise their creativity in designing healthcare delivery models and interventions to meet their local situations and needs [8].

Historically, healthcare in Kenya was initially tax-funded before the introduction of user fees in 1989, and this has remained so in the higher-level facilities under a cost-sharing program [8]. At the same time, the National Health Insurance Fund, introduced in 1966 for formal sector employees, was expanded to provide medical insurance to the informal sector too [8]. Despite this, only about 16% are covered, leaving a significant percentage exposed to the high cost of healthcare, including palliative care. Cancer is currently ranked third in Kenya after infectious and cardiovascular diseases in terms of mortality causes, with an estimated annual mortality of 32,000 cases [9] and over 47,000 new cancer cases each year. Over 80% of these patients are diagnosed in the advanced stages of the disease and in need of palliative care services [7,10]. The increase in the incidence of cancer has been intensified by the HIV/AIDS incidence rate, most notably Kaposi’s sarcoma and the reduced immunity that puts people living with HIV (PLWH) at an increased risk compared to the general population [11,12]. The HIV/AIDS pandemic and the rising prevalence of cancer cases in sub-Saharan Africa, including Kenya, have increased the need for a more developed palliative care structure. However, the concept is neither well understood nor developed, and this has led to palliative care being confined to specialist centers, which are also scarce in Africa. Anecdotal evidence indicates that many people with chronic illnesses avoid visiting hospices due to beliefs that a hospice is a death sentence and hence choose the hospice as the last option when the situation is beyond control. Partly, this could be due to myths surrounding palliative care use [13,14]. The study by Dixe et al. [15]) established a wide gap in knowledge on palliative care and its importance, especially among the public, and as such, has led to underutilization of palliative care services. In addition, other studies have examined factors affecting the utilization of palliative care services. Another study by [16] showed that patients in the lower income quintile were very sensitive to healthcare costs and thus utilized hospice services less frequently compared to those in the upper income quintile. Other studies have pointed out that the costs of healthcare, including palliative services, are a major barrier to accessing these services [17]. In Kenya, the most significant barrier to utilization of healthcare is cost. The 2013 Kenya Household Health Expenditure and Utilization Survey (KHHEUS) showed that 21.4% of those who were sick did not seek healthcare services due to cost despite reporting illness and injury.

While cancer cases continue to increase, the economic burden on patients, caregivers, and society continues to rise significantly [18]. At the same time, many Kenyans reside in rural areas where access to care, specifically palliative care, is a challenge.

There are a myriad of services offered in hospices and other palliative care centers based on several models and settings. The services include pain and symptom control to ensure patients live comfortable lives; spiritual care to cater for the spiritual needs of the dying; homecare; inpatient care; or respite care, depending on the choice of the patient and relatives. Family conferences are also held on a regular basis to update the families of the patients on the patient’s progress. Hospices also act as coordinating sites for the interdisciplinary team that is looking after the patient. In the event of death, the hospice team helps the family cope with and adjust to the loss [11]. When applied, these models help to ensure the accessibility of palliative care to all patients who need this type of care. In Kenya, most of the palliative care centers offer pain management, counseling, home-based care, and cancer education [1].

Despite the existence of more than 70 palliative care centers in the country, utilization of palliative care services remains low in Kenya [19]. Research has shown that palliative care is associated with improved survival and quality of life; however, the factors influencing progressive utilization remain unknown, yet the number of patients in need of the services is significantly growing [20]. In the context of this study, progressive care refers to the continued utilization of palliative care provided to patients who need more monitoring and assessment and whose conditions are not so unstable. This study aimed at bridging the gap within the social context of cancer and palliative care with regards to utilization, as 80% of the patients in Kenya are diagnosed at advanced stages and can only benefit from palliative care services. This study examined the factors associated with progressive utilization of palliative care services by patients with cancer at the Nairobi Hospice, Kenya.

## 2. Methods

### 2.1. Theoretical Framework

Healthcare utilization is associated with the quality and cost of services, individual characteristics, and availability of the services [21]. Behavioral scientists have viewed it as a type of individual behavior that is a function of the individual characteristics, the environmental characteristics, and the interaction of the individual and societal forces [22]. According to Andersen’s conceptual model of health care, access and utilization of health care are related to three main individual factors [23]. They are the predisposing, enabling, and need factors in the PEN model. The predisposing factors refer to socio-cultural characteristics existing prior to the onset of the illness. These include demographic characteristics like age, sex, education, occupation, and social networks, as well as beliefs about health, which include attitudes and values [24]. Age, gender, occupation, and social networks are also key determinants of one’s likelihood of being diagnosed with cancer.

Cancer patients need both diagnostic and palliative care services, and most of these services are expensive, especially where user fees (this refers to partial or complete payment for health services by health consumers in a health facility) are involved. As such, medical insurance and a consistent source of income become necessities [25]. The need factors focus mainly on the health status of the individual. Cancer patients need care, especially those whose diagnoses are made late or whose condition is severe. As such, health workers should be able to advise and direct them to palliative care that fulfills the needs of the individual [20].

### 2.2. Econometric and Model Specification

To analyze the determinants of progressive utilization of palliative care services by cancer patients, the study adopts and modifies the model by Andersen [23]. The model is a multiple regression model and is estimated using a binary probit. In this case, we interpret the dependent variable as the probability of either progressively utilizing palliative care or not, given the explanatory variables. Binary-choice models assume that individuals are faced with a choice between two alternatives, and the choice of any of the two depends on certain factors.
*Y* = *β*o + *βsXs* + *μ*
(1)
where Y is the probability of progressively utilizing palliative care given various explanatory factors, β0 is the constant, βs are the coefficients, Xs are the explanatory variables, and μ is the error term. In this study, we assume that our dependent variable (progressive palliative care utilization) takes the standard normal distribution with a mean of 0 and a variance of 1. Based on this, we estimated the cumulative distribution function (CDF) given the probability distribution function (PDF) as follows:*Pr* (*Y* = 1|*X*) = *Φ* (*Xβ*)(2)
where Pr is the probability of progressive palliative care utilization, Φ is the cumulative distribution function of the standard normal distribution, and β is the parameter to be estimated. The probit model assumes that Y is a normally distributed variable, and therefore Y can be estimated using the probability distribution function given as follows:*pr*(*y* = 1) = *Φ*(*Xβ*) = ∫1√2π*e* − *z*22 *Xβ* − *∞dz*(3)

However, for interpretation purposes, we estimate the marginal effects, which show the change in the probability of the dependent variable given the unit change in the explanatory variable. Therefore, we shall proceed to interpret our results as the probability of progressively utilizing palliative care given a certain factor while holding other factors constant/ceteris paribus. In specifying our model, an assumption is made of a linear relationship between progressive palliative care utilization and the explanatory variables, and we express it as indicated below:*PPCUT* = *β*1 + *β*2*AG* + *β*3*S* + *β*4*ED* + *β*5*MS* + *β*6*ES* + *β*7*MI* + *β*8*REL* + *µi*
where

PPCUT represents the probability of progressive utilization of palliative care by cancer patients. This assumes that patients residing within Nairobi County utilize palliative care continuously compared to those residing outside Nairobi County [25].

AG is the age of the cancer patient seeking palliative care;

S is the gender of the cancer patient;

ED is the education level of the cancer patient;

MS is the marital status of the cancer patient;

ES is the employment status of the cancer patient. It is used as a measure of the economic status of an individual;

MI is the medical insurance status and is used to measure the ability to afford palliative care services;

REL is the religious affiliation of the individual, and it measures the effects of beliefs on palliative care utilization.

μ = the stochastic disturbance term

### 2.3. Data Source and Type

The study employed cross-sectional data on cancer patients seeking palliative care at the Nairobi Hospice. The study site was chosen due to its proximity to the Kenyatta National Hospital (KNH) the country’s main public referral hospital, where most patients with cancer are referred for specialized management. This site was also the first to be established as a palliative care center in the country. The choice of the study period is informed by the period in which the palliative care guidelines came into existence in Kenya. The study is mainly correlational, as it does not seek to analyze the cause-and-effect relationship but aims to look at the relationship between the independent and dependent variables. The study sample comprised 169 cancer patients.

### 2.4. Variables Used and Expected Relationships

The dependent variable is the probability of progressive palliative care utilization, which is denoted by (PPCUT). In this case, progressive utilization assumes that patients from within Nairobi County can continuously attend the palliative care clinics as compared to those living outside the county [25]. The explanatory variables used, and their expected relationships are shown in Table 1.

## 3. Results

### 3.1. Descriptive Statistics

The mean (+SD) age of the 169 participants was 51.89 (+14.67) years, and about 22.9% were within the age group 41–50 years. Most participants were female (77.7%) and unmarried (60.6%); 16.6% were separated, and about 15.4% were windowed. Most of the participants had achieved primary education (35%), while more than half of the participants (58%) had medical insurance. Most of the participants (79.9%) were employed (Table 2).

### 3.2. Econometric Results

Table 3 indicates significant predictors of progressive utilization of palliative care among cancer patients at the Nairobi Hospice. The study used the R statistical tool to analyze the data because of its visualization properties and regression ability. The data set was coded based on the variable under measurement. The results of probit regression showed that the significant predictors of progressive utilization of palliative care are the age of the patient, gender, level of education, employment status, and being a Muslim. The older participants were more likely to utilize palliative care compared to the young participants. Probably this is because the prevalence of cancer is higher among older people compared to young people.

### 3.3. Discussion

The marginal effects were used to calculate the predicted probability of utilizing palliative care services. The results have shown that employment and Muslim religion were significant factors in palliative care utilization. The other factors that were noteworthy at 0.1 significant were age, gender, and education level.

The employment status of the patient was found to significantly influence the progressive utilization of palliative health care. The probability of utilizing palliative care increases by 16.2% among patients with higher employment status. This may be attributed to the availability of finances to facilitate healthcare seeking. The study by Minenhle [25] in KwaZulu Natal found that employed individuals were more likely to utilize health care services compared to unemployed individuals. According to [6], utilization of health care services is robustly associated with employment status, with those who have any employment status seeking healthcare more than the unemployed. This is particularly true for services with user fees. A similar study by [30] found that those permanently employed were less likely to report unmet healthcare needs. Further, it is argued that the utilization of health care services is mostly by affluent members of society who can afford the services offered, as is the case with Nairobi Hospice.

Our results also found that being a Muslim was an important predictor of the progressive use of palliative care services. Previous research has shown that being spiritual is a sense of peace and determination inside our being, and religion is a construct of human making to express spirituality [31]. Spirituality and religion play an essential role in medical healing, especially when patients face the crisis of advanced illnesses towards the end of life. A study by [32] in Indonesia concluded that spirituality/religious aspects are important in palliative care provision. Glaser’s survey done for 16 countries revealed a positive association between religious beliefs in a society and the utilization of medical facilities [5]. There were also fundamental differences among societies in the extent to which permission was granted for a person to be hospitalized. These cultural differences also determined how people interacted in the community, family structure, and authority. This, in turn, affected the sources of medical care.

Older age was also associated with increased use of palliative care; indeed, the results showed that older patients increased the probability of using palliative care by 0.004. The probability of utilizing palliative care significantly increases by 0.004 among older patients. A similar study by Jason [33] showed that utilization increased with age among elderly patients. Why does the age of the patient have such a strong association with the use of palliative care? One explanation for the increased use of palliative care by older patients is the increased prevalence of chronic illnesses like cancer among the elderly and the fact that their needs are catered for more in the hospice than elsewhere. The high proportion of older cancer patients dying in hospices may also reflect an effective approach to meeting the needs of these patients. It is also likely that younger patients are served late by specialized palliative care services, and this may correlate with their higher probability of experiencing aggressive hospital treatment, which can delay referral to specialized palliative care services [34].

Our study found that being female suggestively increases the probability of progressively using palliative care by 14% compared to male patients. Earlier research reported that females had higher rates of cancer-related morbidity compared to males and were more likely to seek palliative care than males. This is also attributed to both biological and social differences between the two genders [35]. Women were also found to view palliative care as more efficacious relative to men [36].

The level of education was found to considerably influence progressive palliative health care utilization. A higher level of education increases the utilization of palliative care progressively by 5.6%. Our results support previous studies that found that higher levels of education improve the probability of seeking health care [14]. One explanation for this finding is that patients with higher education are better able to identify and articulate their healthcare needs and thus seek them. Furthermore, education has been shown to often influence the occupation of an individual, and this will directly translate to the income earned and the ability to afford healthcare services [22].

Contrary to our expectations, medical insurance was found to reduce the progressive utilization of palliative care services by 4.1%. While most of the patients had statutory National Health Insurance Fund (NHIF) coverage, it did not include palliative care in its benefit package and therefore may not be adequate to meet the cost of the palliative care offered. At the same time, a lack of palliative care benefits may have discouraged their use. An example is Ghana, where the National Health Insurance covers only pro-poor health financing interventions. The only cancers covered by the scheme are breast and cervical cancers, and the scheme does not clearly cover palliative care.

## 4. Conclusions

This cross-sectional survey examined the factors associated with cancer patients’ utilization of palliative care at the Nairobi hospice. The study found significant positive predictors of palliative care utilization in employment status and religion. This study can guide health policymakers in growing the rate of palliative care utilization in the country. The suggested policies include ensuring that the universal health care policy includes conditions that require palliative care. This will enable managers in hospice facilities to attend to patients irrespective of financial requirements for the treatment of a given ailment. Such a policy will also ensure that the problems often induced by the financial incapacitations of patients will be removed. Furthermore, to increase patients’ access to palliative care services, efforts are required to enhance the support and education given to patients and their families. Given that religion, especially being a Muslim, significantly influences the progressive utilization of palliative care services, it is reasonable for policymakers to understand and apply religious and spiritual aspects to the patients’ holistic health in palliative care treatment.

## Figures and Tables

**Table 1 ijerph-20-06871-t001:** Variables used and expected relationships.

Variable	Measure	Expected Relationship with PCUT
Progressive Palliative care utilization (PPCUT)	We assumed that patients residing within Nairobi County utilize palliative care progressively compared to those residing outside Nairobi County due to accessibility of the service. It takes a value of 1 if the residence is in Nairobi County and 0 otherwise.	
Age (AG)	Continuous variable	Positive; increase in age leads to an increase in utilization [26]
Sex (S)	Dichotomous variable; Male = 0 and female = 1	Positive; Females utilize more than males [2]
Education level (ED)	Discrete variable; None = 0, Adult education = 1, Primary = 2, Secondary = 3, post-secondary = 4	Positive; increase in the number of completed school years leads to increase in utilization of palliative care services [27]
Marital status (MS)	Discrete variable; classified as married = 1, widowed = 2, single = 3 or separated = 4	Marriage increases utilization [28]
Employment status (ES)	Dichotomous variable; employed = 1, unemployed = 0	Positive; employment increases utilization [17]
Medical insurance (MI)	Dichotomous variable; insured = 1 uninsured = 0	Positive; Having a medical insurance increases utilization [29]
Religion (REL)	Discrete variable; None = 0, Christian = 1, Muslim = 2 and Hindu = 3	Indeterminate, since some religion advocate for general utilization of health care services while others oppose [7]

**Table 2 ijerph-20-06871-t002:** Socio-demographic characteristics of the participants.

Variable	Observations	Mean	Std. Dev
Progressive Utilization of palliative care	166	0.2710843	0.4458645
Age	167	54.07784	17.52141
		**Absolute numbers**	**%**
Gender (Female = 1)	168	102	61
Married	168	105	62
Widowed	168	26	15
Single	168	29	17
Separated	168	8	4
No Education	168	22	13
Adult Education	168	05	3
Primary Education	168	60	35
Secondary Education	168	43	25
Post-Secondary Education	168	33	20
Employed	164	131	79
Medical Insurance	165	97	58
No religion	168	8	4
Christian	168	151	89
Muslim	168	5	2
Hindu	168	2	1

**Table 3 ijerph-20-06871-t003:** Probit regression analysis to determine factors associated with progressive utilization of palliative care (marginal effects).

Progressive Palliative Care	Marginal Effects	Std. Err.	Z	P > z	[95% Conf. Interval]	
Age	0.0037763	0.0020772	1.82	0.069	−0.0002949	0.007847
Gender	0.1407649	0.0763362	1.84	0.065	−0.0088513	0.290381
Single	0.1386041	0.0947258	1.46	0.143	−0.0470552	0.324263
Widowed	−0.0445677	0.0968539	−0.46	0.645	−0.2343979	0.145262
Education	0.0567334	0.0294349	1.93	0.054	−0.0009579	0.114424
Employment	0.162673	0.0718241	2.26	0.024 *	0.0219003	0.303445
Medical insurance	−0.041035	0.0692198	−0.59	0.553	−0.1767032	0.094633
No religion	−0.2021024	0.1842968	−1.10	0.273	−0.5633175	0.159112
Muslim	0.4151239	0.2052956	2.02	0.043 *	0.012752	0.817495

* Significant standard errors at 5%.

## Data Availability

The data in this study are available upon request from the corresponding authors. The data is not publicly available due to privacy issues.

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
