# Peer review of "Factors Influencing Progressive Utilization of Palliative Care Services among Cancer Patients in Kenya: The Case of Nairobi Hospice"

_ijerph, 2023, doi:10.3390/ijerph20196871_

Round 1
Reviewer 1 Report (Previous Reviewer 2)
1. It is not clear whether tool for statistical calculation was used, R, SAS or other application ? If you used original program, describe how to validate the calculation.
2. In Table 3, 3-4 significant digit should be limited.
3. Equation 3 and PCUT should be improved for completing a final manuscript.
Author Response
Dear Reviewer
Thank you for the valuable comments on our manuscript. These helped to improve the quality of the article. Attached are the responses to the comments provided.
Kind regards
Catherine

This manuscript is a resubmission of an earlier submission. The following is a list of the peer review reports and author responses from that submission.
Round 1
Reviewer 1 Report
Thanks for the paper. The authors studied the PEN (predisposition, enabling and need) factors related to progressive utilization of palliative care services of cancer patients in Kenya. The study was a cross-sectional survey and probit regression model was used to understand the factors contributing to the use of progressive palliative care. There are a few issues for the authors to clarify.
1 This journal is for international readers, so it would be helpful for the authors to introduce the Kenyan healthcare system, particularly the funding and consumer co-payment regime related to cancer treatment and palliative care.
2 Study subjects selection: it is not clear to me how the study subjects were selected. The authors said there were 169 cancer patients (line 161), and then said 394 participants (line 171). I'd like to know clearly how many patients used the progressive palliative care and how many did not. This part should be enhanced rather than the probit regression model (which can be simplified).
3 Study period: it should be described clearly before lines 158-159. The policy implications of this study is not so significant given the study used the data almost 10 years ago. There might have been subsequent improvements to the use of palliative care since then in Kenya.
4 Probit regression model was used to study the effects of the PEN factors on the use of progressive palliative care. PCUT was supposed to be the dependent variable. In Table 1, what's the relationship between residence in Nairobi county and PCUT?
5 Table 2: This table needs to be redone. For continuous variables, mean and SD may be appropriate (if they follow normal distribution); for categorical variables like gender, marriage, education level, employment, medical insurance and religion, absolute number and proportion should be reported.
6 Table 3: it would be helpful to include the reference category for each independent variable. In addition, the authors should indicate the significance level in the methods. Actually, there were only two variables significant at 0.05 level: employment and religion (Muslim). Because of these findings, the discussions need to be adjusted, e.g. regarding medical insurance (which was not a significant factor for use of palliative care). The stars * or ** should be used to label the P>Z rather than the standard errors.
7 Words missing for the definition of progressive palliative care (lines 83-84).
8 Line 193: "determinants" is a strong word for a statistical model using a cross-sectional study design, so should be avoided.
9 Line 195: 0.004%? or 0.004?
Reviewer 2 Report
1. The manuscript examined the factors associated with progressive utilization palliative care service in Kenya. Palliative care service differs between developed and developing countries. In Introduction, this point should be discussed by reviewing some papers.
2. Statistics uses regression analysis of a binary probit model. What is a statistical tool like R, SAS ? If you use an original program, you should validate your statistical calculation.
3. PCUT in statistical analysis uses binary data that show Nairobi residence or not. This would be a key problem in this analysis. The objective variable in this analysis shows Nairobi residence is 1 otherwise 0. This analysis only examines trends such as age and education of Nairobi residents.
4. Table 3 shows the result of the statistical analysis. This shows the variables of Empolyment ad Muslim provided statistical significance according to the confidence interval.
5. References should be improved to be cited exactly.
6. Equation formula is not completed, should be improved.